# Imitative Reinforcement Learning Fusing Mask R-CNN Perception Algorithms

**Lei He** [1] **, Jian Ou** [1,*]**, Mingyue Ba** [2]**, Guohong Deng** [1] **and Echuan Yang** [3]

1   Key Laboratory of Advanced Manufacturing Technology for Auto Parts, Ministry of Education, Chongqing University of Technology, Chongqing 401320, China
2   Chongqing Chang'an Automobile Co., Chongqing 400023, China
3   College of Mechanical Engineering, Chongqing University of Technology, Chongqing 401320, China
*   Correspondence: oujian@cqut.edu.cn

**Abstract:** Autonomous urban driving navigation is still an open problem and has ample room for improvement in unknown complex environments. This paper proposes an end-to-end autonomous driving approach that combines Conditional Imitation Learning (CIL), Mask R-CNN with DDPG. In the first stage, data acquisition is first performed by using CARLA, a high-fidelity simulation software. Data collected by CARLA is used to train the Mask R-CNN network, which is used for object detection and segmentation. The segmented images are transformed into the backbone of CIL to perform supervised Imitation Learning (IL). DDPG means using Reinforcement Learning for further training in the second stage, which shares the learned weights from the pre-trained CIL model. The combination of the two methods is an innovative way of considering. The benefit is that it is possible to speed up training considerably and obtain super-high levels of performance beyond humans. We conduct experiments on the CARLA driving benchmark of urban driving. In the final experiments, our algorithm outperforms the original MP by 30%, CIL by 33%, and CIRL by 10% in the most difficult tasks, dynamic navigation tasks, and in new environments and new weather, demonstrating that the two-stage framework proposed in this paper shows remarkable generalization capability in unknown environments on navigation tasks.

**Keywords:** Mask R-CNN; DDPG; conditional imitation learning

## 1. Introduction

Autonomous driving has made significant progress in the last decade. To date, there are two main paradigms for vision-based autonomous driving systems: the mediated perception approach, which makes driving decisions by parsing the entire scene, and the behavioral reflection approach, which maps input images directly to driving actions via a regulator [1]. The behavioral reflection approach, also known as the end-to-end approach, has performed reasonably well over the last five years. Imitation learning for end-to-end autonomous driving has attracted academic attention.

There are two trends in training research for end-to-end driving models. One is reinforcement learning. Our knowledge, however, indicates that many current reinforcement learning-based driving methods are based on trial-and-error reinforcement learning. These methods are difficult to apply to the real world because the training process is not safe. Imitation learning is the second method. Despite the ease of understanding and implementing imitation learning, policies that only learn from expert demonstrations may be unable to recover from mistakes as a result of a lack of a recovery process. For example, the DeepMind parkour paper [2] used 6400 CPU hours to achieve the results in the paper. Sample Efficiency is not really noticeable on these platforms, it can be allowed to run in this virtual environment, but in realistic scenarios such as robotic tasks, it poses a major obstacle, after all; it is costly to keep a robot running for many hours in reality.

However, refs. [3–5] shows that when only a single RGB image is used as input, only the presence or absence of obstacles can be detected, but the exact location of the obstacles is unknown, especially in some extreme weather such as rain, snow, and fog which can produce noise on the camera image and ultimately bring about inadequate acquisition of information about the driving environment around the vehicle. To compensate for these deficiencies, the acquired images are semantically segmented. In computer vision, image segmentation differs from classification and target detection in that it is usually a low-level or pixel-level vision task, as the spatial information of the image is important for semantically segmenting different regions. Segmentation aims to extract meaningful information for analysis.

Xi Liang [6] proposes a feature fusion and scaling-based single-shot detector (FS-SSD) for small object detection in UAV images. Six experiments were conducted on the PASCAL VOC and two UAV image datasets. The experimental results show that the proposed method can achieve comparable detection speed, but its accuracy is better than the six state-of-the-art methods. In this work [7], the YOLO V3 is used to detect the network of objects in the picture. In addition, a steering angle circuit has been designed and implemented to measure the direction of the car. The steering angle measurements are used with object detection (vehicles and pedestrians) to warn when these objects are close to the driving car (10 m). Once an object has been detected using the YOLO V3, the height of the object is used to measure the distance of the detected object. Affordable and low cost while achieving positive and competitive results, this system can be used at night and in dark environments.

Across all imitative learning methods, performance dropped by at least 2 when transitioning to challenging navigation tasks. Possibly this is due to the model not being able to generalize to new towns using different textures and 3D models. Overall, the experimental results of the method emphasize the importance of generalization for learning-based sensorimotor control methods [8]. As well as this, imitation learning's ability to generalize to complex conditions and unseen environments is dependent on the training data, so there is room for improvement [9]. In summary, robustness to extreme driving conditions and generalization performance to a variety of environments are the two main challenges for autonomous urban driving.

This paper addresses severe weather conditions by combining reinforcement learning with imitation learning and fusing Mask R-CNN algorithms [10]. The problem of generalization in imitation learning exists, and it can be addressed by reinforcement learning using semantic segmentation of images and using image enhancement. A pre-trained imitation learning model is used to initialize the participant network by sharing weights with the participant network. A reward function can be used to interact with the environment and then receive a reward based on how the interaction is performed.

Unlike the data-driven process of imitation learning, reinforcement learning RL is a self-learning algorithm that allows self-driving cars to perfect their driving performance through repeated trials without depending on humanly set rules or manual driving data. The comprehensive model in this paper is based on DDPG, an actor-criticism algorithm based on replayed memory. [11] In this paper, the actor network is initialized by sharing weights with an imitation learning pre-trained model and optimizing it according to a reward function. By interacting with the environment and receiving rewards, the driving agent can learn a driving strategy that performs well in dynamic navigation tasks. An abbreviated framework for this paper is shown in Figure 1.

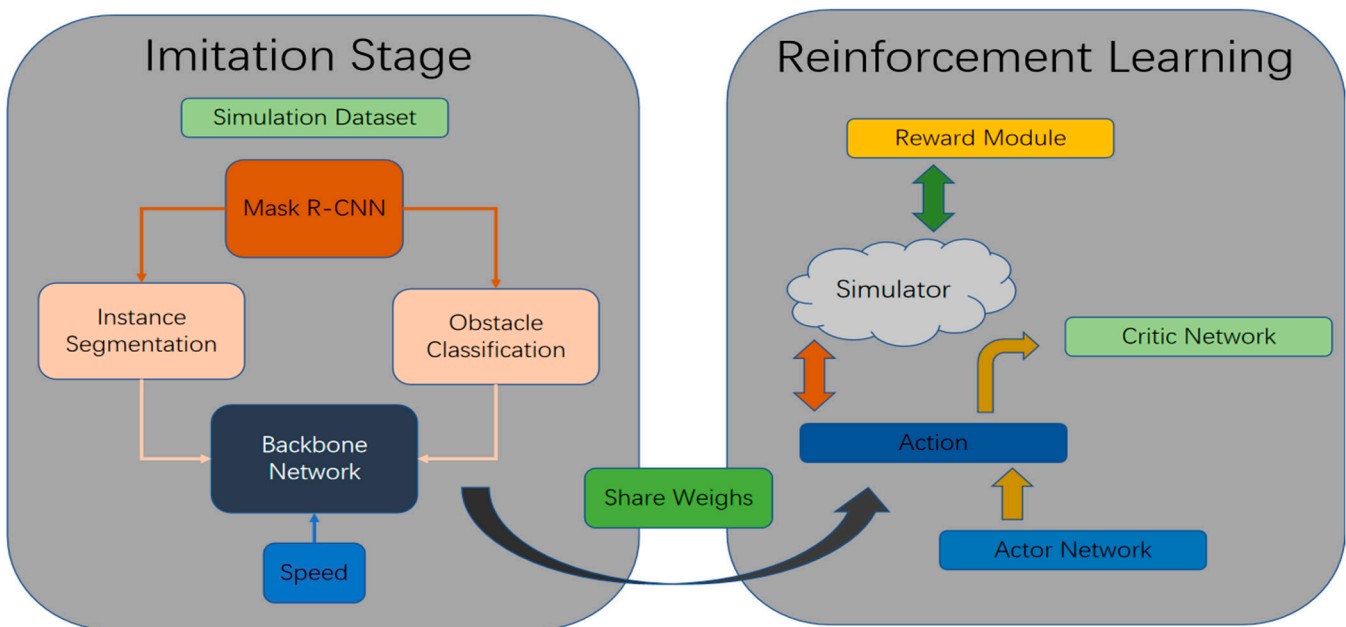

**Figure 1.** Structural framework of this paper.

The related issues to be addressed in this paper are as follows:

It is not hard to have a reward function. However, designing a reward function that allows the agent to learn the desired behavior is difficult. In short, sparse rewards can make learning difficult for the agent, but if a lot of manually designed rewards are added by the individual, the agent may learn unintended behaviors if the rewards are poorly designed. Current reinforcement learning algorithms are opaque, and in most cases, we have only high-level intuition about what a reinforcement learning algorithm can learn and how it will work. For most problems, we want the algorithm to be predictable and interpretable. The least explanatory and predictable approach is a large neural network that learns the desired knowledge from scratch, given only low-level reward signals or an environmental model (like AlphaGo Zero).

In addition to this, the paper will also consider bad weather [12–14] bad weather. These papers suggest that there is still room for improvement, and it has not yet been determined whether the most promising robustness enhancement techniques require structural modifications, data enhancement schemes, modifications to the loss function, or a combination of these. The relevant solution in this paper is to use Mask-RCNN [10] for multiple processing of obstacles, including obstacle recognition and instance segmentation to process obstacles to obtain information about them and add a pre-trained model for imitation learning to add robustness.

The original innovations and contributions of this work are reflected in the following aspects:

1. This paper proposes a two-stage framework called CIL-DDPG, which combines reinforcement learning and imitation learning using obstacle position information as additional input, and Mask-RCNN [10] is used for image segmentation. The collection of CIL and DDPG is an innovative development; previous releases have used the two methods separately as one development algorithm; secondly, this paper has made innovations such as obstacle information fusion in CIL.

2. A new reward function is designed to learn an alternative autonomous driving strategy in a dynamic scenario. Extensive experiments on the CARLA simulator benchmark show that the work in this paper enables the network to overcome the effects of image noise.

To better explain the content of this paper, the rest of the paper is organized as follows. Section 2 discusses some important related works, while Section 3 discusses the Mask-

RCNN concepts. Section 4 discusses the concepts of conditional imitation learning. The reinforcement learning algorithm is described in Section 5. Section 6 contains simulation experiments. Finally, findings are offered, along with potential future study areas.

## 2. Related Work

Deep learning-based image segmentation algorithms, such as VGGNet [15] and ResNet [16], are extremely well-preferred. The image pixels are labeled, so each pixel shares certain features, such as color, intensity, and texture. To date, these two networks still have an extremely high dominance in the field of feature extraction.

Long J et al. [17] presented FCN networks at CVPR in 2015, proposing full convention­alization of neural networks, using convolutional layers instead of the final fully connected layer to complete the segmentation task. Many network models still borrow the structure of FCN networks to this day.

Zhang et al. [18] proposed an algorithm called Mask Scoring R-CNN that was used for traffic monitoring to obtain comprehensive vehicle information such as vehicle type, speed, length, current driving lane, etc. Eventually, the average recognition accuracy for the model and the number of axles was above 97% and 88%, respectively.

Due to an evolution based on Mask R-CNN networks, the combination of ResNeXt-101+FPN can be said to be the best feature learning now. The specific improvement includes the segmentation loss, varying from the original FCIS polynomial cross-entropy based on single-pixel softmax to single-pixel signed binary cross-entropy. ROIAlign, an interpolation of feature maps, solves the misalignment problem. Therefore, in this paper, Mask R-CNN is used.

In 2016, Bojarski et al. [19] trained a CNN to drive autonomously on different types of roads and achieved over 10 miles of lane-keeping. The network achieved an autonomous driving rate of 98% through real-world testing.

Another deep CNN, PilotNet, was trained using road images from a single front-facing camera paired with driver-generated steering angles captured inside the cabin [20]. A drawback of the above work is that their performance comes from a large amount of training data with manual markers.

Hesham et al. [21] consider that most existing solutions only consider visual camera frames. Therefore, this work, proposed a convolutional long short-term memory recurrent neural network (C-LSTM), which is an end-to-end approach to learning visual and dynamic time-dependent driving. Although their study ultimately achieved good performance, the nature of the vision-based study failed to avoid the effects of the weather environment.

Codevlla et al. [22] used high-level commands as additional input to build a con­ditional imitation learning (CIL) model. Another end-to-end example is the ICCV 2019 Learning to Drive Challenge, where Columbia University's deep learning team rounds out the top two. The fusion of data from camera sensors and visual maps resulted in significant performance improvements. While these end-to-end approaches have proven to perform well in real-world experiments, the robustness and generality need to be improved.

Wang et al. [23] proposed a new navigation command that does not require human involvement and a new model structure, the angular branching network. Furthermore, in addition to segmentation information, depth information can also improve the performance of the driving model. They conducted experiments in both qualitative and quantitative evaluation to show the effectiveness of the model.

In recent years, deep reinforcement learning (DRL) methods for decision-making in self-driving cars have been increasingly researched. One reason is its great success on many artificial intelligence tasks; another well-known shortcut of imitation learning is the weak generalization performance and the risk of overfitting the training data.

However, decision-making in autonomous driving remains a challenge. Reinforce­ment learning (RL) has been used to obtain correct behavior in uncertain environments automatically, but it cannot guarantee the performance of the final policy.

Maxime et al. [24] propose a general approach to enhance the probabilistic guarantees of RL agents. An exploration policy is derived before training, constraining the agent from choosing among actions that satisfy the desired probability specification in a linear time logic (LTL) representation. Reducing the search space can simplify reward design.

Jianyu Chen et al. [25] proposed a framework that allows model-free deep reinforcement learning to be applied to challenging urban autonomous driving scenarios. A bird's-eye view input representation was designed to reduce sample complexity, and visual coding was used to capture low-dimensional latent states. While the adaptation method outperforms the baseline, it does not solve the task perfectly. By using reinforcement learning (RL), strategies can be learned and improved automatically without any manual design. However, current RL methods are usually unsuitable for complex urban scenarios. Furthermore, to perform more complex autopilot tasks, we would design a more efficient reward function [26].

This section presents some related work that combines reinforcement learning with imitation learning. The core idea is that the agent can learn quantitative parameters from the image data, which can represent information about the state of the road. These parameters are then used to control the vehicle.

Over the past decade of its role, researchers have achieved good results with end-to-end approaches. However, the approach is generally poorly adapted to the environment, especially for dynamic traffic environments. Image information is acquired by using the front-facing camera of one's vehicle, which is then fed into a carefully designed convolutional network to extract features that represent the current state of the vehicle environment. The feature information is then fed back into the reinforcement learning framework for learning. Finally, the reinforcement learning model directly outputs the amount of steering, throttle, etc., that the vehicle will control at the next moment in time.

Mingxing Peng et al. [27] proposed a two-stage framework called IPP-RL. In their IPP model, the visual information captured by the camera is compensated by the steering angle calculated by a pure tracking algorithm. It can therefore operate well in adverse weather conditions. However, with reliance on visual information, there may be obstacle misdetection, and the approach is still inadequate in more challenging and complex driving conditions where vehicles are unable to grasp safe distances for collisions.

Xiaodan Liang et al. [28] proposed a general and rule-based Controllable Imitative Reinforcement Learning (CIRL). To alleviate the low exploration efficiency of large continuous action spaces, the CIRL initializes the pre-trained model weights of the actor network through imitation learning. Furthermore, CIRL also proposes adaptive strategies and steering angle reward functions for different control signals (i.e., following, straight ahead, right turn, left turn) to improve the model's ability to handle varying situations. The heavyweight references relevant to this study are shown in Table 1.

**Table 1.** Summary table of algorithms related to this article.

| Algorithm Name and Reference | Brief Methodology | Highlights | Limitations |
|---|---|---|---|
| Imitation learning fusing Pure-Pursuit Reinforcement Learning) | In their IPP model, the visual information captured by the camera is compensated by the steering angle calculated by a pure tracking algorithm. | It is robust to lousy weather conditions and shows remarkable generalization capability in unknown environments on a navigation task. | IPP-RL uses Pure-Pursuit to increase computing power and does not meet real-time requirements; in addition to this, it results in a complex model with reduced robustness. |
| Controllable Imitative Reinforcement Learning (CIRL) | The CIRL initializes the pre-trained model weights of the actor network through imitation learning. | CIRL also proposes adaptive strategies and steering angle reward functions for different control signals (i.e., following, straight ahead, right turn, left turn) to improve the model's ability to handle varying situations. | CIRL does not perform image classification and image enhancement, and in practice, some false detections occur. |

## 3. Mask R-CNN

### 3.1. Structure of Mask R-CNN

Mask-RCNN [10] is the best paper of ICCV2017. Mask-RCNN is an improvement on Faster-RCNN by adding a fully connected segmentation sub-network. The model changes from two tasks (classification + regression) to three tasks (classification + regression + segmentation). The structure allows the semantic segmentation of the target while implementing object detection. The detection is first done on the image to find out the ROIs in the image, pixel correction is performed for each ROI using ROIAlign, and then the prediction of the different instance belonging classification is performed for each ROI using the designed FCN framework to obtain the image instance segmentation result finally. Mask R-CNN is a two-stage framework, where the first stage scans the image and generates proposals (proposals, i.e., regions that are likely to contain a target), and the second stage classifies the proposals and generates bounding boxes and masks. The workflow of Mask-RCNN is shown in Figure 2.

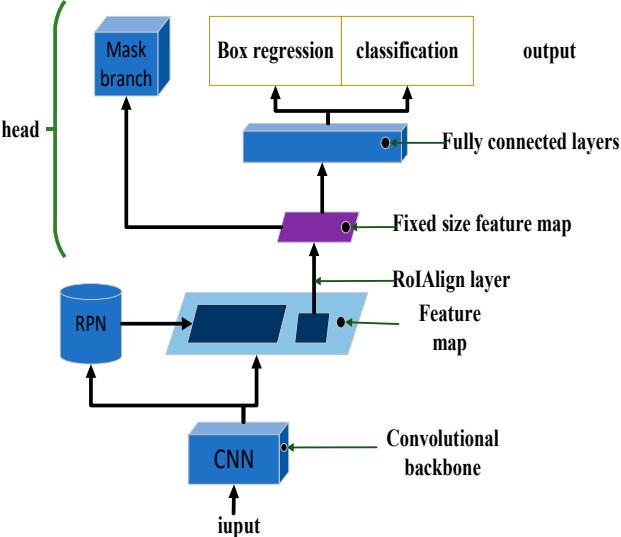

**Figure 2.** The workflow of Mask-RCNN.

Algorithm 1 describes the Mask-RCNN pseudo-code flow.

---

**Algorithm 1.** Mask-RCNN

---

**Input:** the RGB images
**Output:** Image with category, mask and bounding box
**Repeat:** until there is no rgb image input
    **Step 1:** The RGB images are fed into ResNet101 for feature fusion;
    **Step 2:** Then two feature maps are generated as rpn_feature_maps and mrcnn_feature_maps;
    **Step 3:** Different sizes of rpn_feature_maps are sent to the RPN in the feature extraction phase;
**Step 4:** After the RPN, the rpn_class, rpn_box and the anchor generator generated from the anchors, finally go to the Proposal Layer;
**Step 5:** Mapping proposals of mrcnn_class, mrcnn_bboxes and iuput_image_meta to the final layer of the DetectionTargetLayer;
    **Step 6:** Generating a fixed-size feature map for each RoI using an RoI Align layer;
    **Step 7:** The detections are combined with mrcnn_feature_maps to fpn_mask_graph;
    **Step 8:** Final generation of mrcnn_masks.
**End repeat**

---

As can be seen, Mask R-CNN is trained by sending feature maps of different sizes to the RPN in the feature extraction phase. The choice of multiple feature maps was chosen because it is known that there are different sizes of targets on the graph. The advantage is that when the targets are large, it is good to use low-resolution feature maps to detect large targets; correspondingly, when the targets are small, it is good to use the high resolution to detect small targets. This is the reason why the backbone chose resnet + fpn.

After RPN, a large number of candidate regions are generated, which need to be cut out using ROI on several feature maps of different sizes, i.e., the target region. The target regions are then fed into ROIAlign (faster is ROIPooling) for subsequent classification and regression.

### 3.2. Dataset Acquisition

It is recommended that standardized fonts such as Times New Roman and Arial are used with a font size no smaller than 10 pt. In order to compare CIRL and CIL in the later experimental results, the paper uses the same experimental setup as [7] to validate the effectiveness of our imitation reinforcement learning.

The information obtained by the sensors is from the forward-facing image camera, the velocity measurements from the simulator, and the navigation planner by the generated control commands. In this paper, the CARLA simulator is used, as in [16]. The dataset includes RGB images, controls, and measurements for each step.

The dataset was collected in the CARLA simulator, using the specified keys on the keyboard to control the car to collect images and labels in the city as a sample set.

To obtain more image information, the image size was set to 800 * 600, the number of vehicles to 15, the number of pedestrians to 30, and the FPS to 10. A total of approximately 700,000 images were acquired, including both the original RGB images and the converted semantic segmentation images.

While the images were captured, the labels and control information was also saved as CSV tables, with each row containing the vehicle position coordinates, vehicle pose, control information, etc.

There are 28 labels corresponding to the images according to the CARLA default acquisition method, but only five labels are used: speed, steer, throttle, brake, and high-level Commands, as shown in Table 2. Steering range at [−1.0–1.0].−1.0 means full left rudder, 1.0 means full right rudder; throttle range at [0.0–1.0], 0 means no throttle, 1 means maximum throttle; brake range at [0.0–1.0],0 means no brake, 1 means maximum brake.

**Table 2.** Tag value information.

| Serial No. | Control Volume | Type | Description |
|:---:|:---:|:---:|:---:|
| 1 | Speed | float | - |
| 2 | Steer | float | [−1.0, 1.0] |
| 3 | Throttle | float | [0.0, 1.0] |
| 4 | Brake | float | [0.0, 1.0] |
| 5 | High-level command | int | (2 Follow lane, 3 Left, 4 Right, 5 Straight) |

### 3.3. Image Enhancement

The size of the images captured by the CARLA simulator is 800 * 600, which is slow and prone to over-fitting if used directly for training, so some processing is required first.

Firstly, the image was resized by cropping off the top part of the sky and the bottom part of the car hood, leaving an 800 * 352 image, and then it was subsampled twice to reduce the size by 200 * 88.

There are three reasons to explain this: firstly, due to the limitations of our equipment, the video memory is too small to process large images, so reducing the size can improve the speed of image processing; secondly, smaller images can use smaller convolutional kernels to reduce the number of operations, which has been commonly used since VGG; thirdly, a large image with fewer convolutional layers will lead to a higher dimension of Flatten, and the final output will have a huge number of parameters, resulting in a complex model, while smaller image inputs can simplify the model, avoiding the problem of overfitting.

As the captured images are too homogeneous, augmentation is required to increase the data sample and its diversity.

The typical image enhancement method is to flip the image, adjust the brightness, add shadows and move the image. The Figure 3 below shows each enhancement method's before and after image comparison.

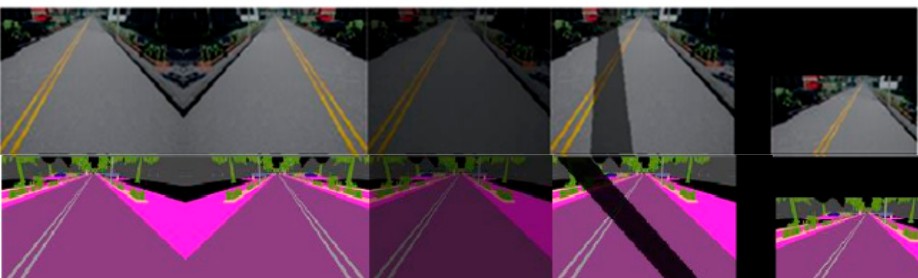

**Figure 3.** The effect of image enhancement.

## 4. Imitation Learning

### 4.1. Conditional Imitation Learning

In this work, the structure of the model, the velocity module, and other settings are consistent with CIL [3]. The biggest difference is the use of the output of Mask-RCNN.

Two fully connected layers connect all backbone speed modules. Each contains 512 units in the image module and 128 units in the speed module.

A fully connected layer connects the backbone with 512 units, and velocity modules are composed of a fully connected layer with 512 units. Each branch is trained separately using a high-level multi-branching-based mechanism command. Online enhancement and method enhancement during training of the data network is performed as in CIL [3].

The image size is 200 * 88 * M, with M = 3 representing the input RGB image; with M = 1, the input is a semantically segmented image.

The input is normalized, speeding up the gradient descent to find the optimal solution and accelerating the convergence to transform the pixel values between [0, 1]. Depending on the backbone, the input M is adjusted to achieve different inputs to the model. As shown in Figure 4.

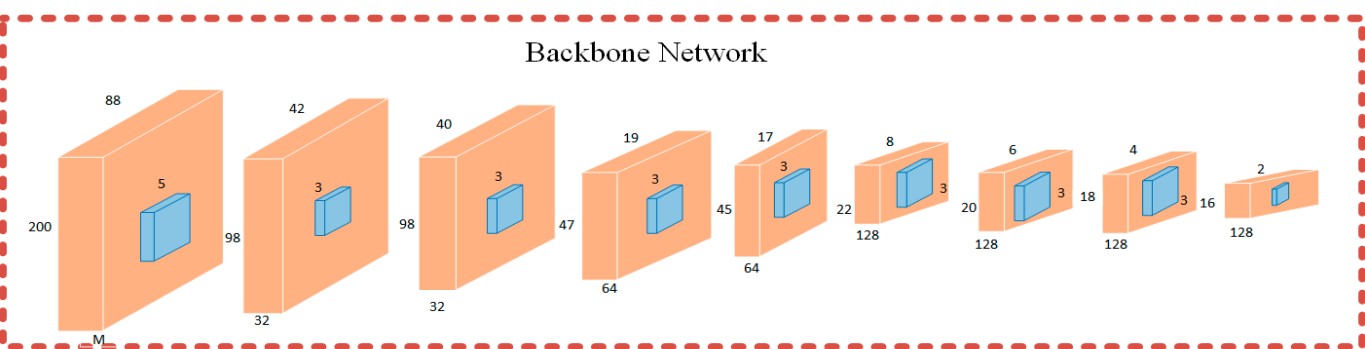

**Figure 4.** Conditional imitation learning network structure.

In the training phase, the position of each step is defined as the planning path, with one point per 0.4 m scattered path. In the test phase, the paths are planned by the planner in CARLA.

The dataset can then be interpreted as:

$$D = \{\langle o_i, s_i, c_i, p_i, a_i \rangle\} \tag{1}$$

where: $o_i$ is the sensor data observation, which is referenced to RGB image information or semantic segmentation images in this paper; $s_i$ is the vehicle speed; $c_i$ is the advanced

command; $p_i$ is the steering result; $a_i$ is the vehicle ground truth action, including steering angle, acceleration, and braking for each step.

The action predicted by the network is defined as follows

$$a'_i = \pi(o_i, s_i, c_i, p_i) \tag{2}$$

Using the L2 loss function:

$$\mathcal{L}(a'_i, a_i) = \left|\left|a'^{s}_i - a^{s}_i\right|\right|_2 + \left|\left|a'^{a}_i - a^{a}_i\right|\right|_2 + \left|\left|a'^{b}_i - a^{b}_i\right|\right|^2 \tag{3}$$

where: $a^{s}_i$ is the steering angle; $a^{a}_i$ is the acceleration; $a^{b}_i$ is the braking action;

The network is trained to minimize the gap between the predicted steering commands and the underlying facts. In practice, the best parameters $\theta'$ are obtained by minimizing the loss of;

$$\mathcal{L}(a'_i, a_i) = \left|\left|a'^{s}_i - a^{s}_i\right|\right|_2 + \left|\left|a'^{a}_i - a^{a}_i\right|\right|_2 + \left|\left|a'^{b}_i - a^{b}_i\right|\right|^2 \tag{4}$$

*4.2. Training and Validation*

The inputs include the original RGB image and the semantically segmented image, as well as control information (i.e., measurements). The image is subjected to information feature extraction by a convolutional neural network, which outputs a predicted velocity value.

The predicted velocity values are fused with the control information extracted from the fully connected network, and the model outputs the predicted action values combined with the high-level control commands, which give more accurate results for each branch of the prediction.

As can be seen in Figure 5 below, the loss profile of the model tends to decrease as the number of iteration steps increases. Although there is some jitter in all the intermediate training losses, they eventually level off. The loss profile no longer decreases, indicating that the network has converged. At this point, the model has reached the optimal result, and the validation loss is slightly lower than the training loss in the early stage, indicating that the model does not appear to be overfitted during the training process, and the training result is good.

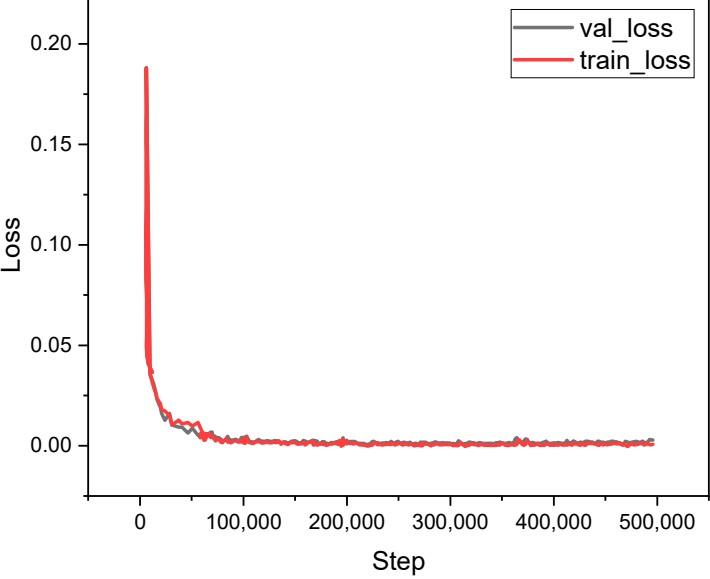

**Figure 5.** Comparison graph of training loss and validation loss.

## 5. Reinforcement Learning

*5.1. Markov Decision Process (MDP)*

By interacting with the car simulator, the agent can optimize according to the reward signals provided by the environment without human intervention, which can be defined as a Markov Decision Process (MDP).

In an autonomous driving scenario, the MDP is defined by a tuple <I,C,S,A,R,P,λ>. In an autonomous driving scenario, the MDP is defined by a tuple that consists of a set of states $O$, defined by observed frames $I$, velocities $S$, control commands $C$; a set of actions, a reward function, a transition function $R(s_t, a_t)$, $P(o'|o,a)$ and a discount factor $\gamma$.

After performing the action and interacting with the environment, the agent receives a reward and arrives at a new state according to a probability distribution.

In each state, the client subject performs an action $a \in A$. After taking that action and interacting with the environment, the agent receives a reward and arrives at a new state according to a probability distribution. To make driving strategies more realistic, the vehicle must follow the path generated by the topology planner to reach the intended goal. New observations $o'$ are updated by simulator observations and a series of commands towards the goal. The event terminates when the vehicle reaches the target, collides with an obstacle, or when the time budget is exhausted.

Deterministic and static policies $\pi$ specify the actions that the agent will take in each state given. The goal of the driving agent is to find policies $\pi$ that map states to actions that maximize the total expected discounted payoff. Thus, this can be learned by using an action-value function: $Q^\pi(o,a) = E^\pi \left[ \sum\limits_{t=0}^{+\infty} \gamma^t R(o_t, a_t) \right]$ where is the expectation $E^\pi$ of the distribution of allowable trajectories $(o_0, a_0, \ldots, o_t, a_t)$ by executing the policies $\pi$ sequentially over some time.

As the autonomous driving system needs to predict continuous movements (steering angle, braking, and acceleration), we use an actor-critic network for the continuous control problem, where both actor and critic are parameterized by a deep network.

In this work, we used the deep deterministic policy gradient DDPG, a model-free algorithm based on actor critique that can operate on a continuous action space. The DDPG algorithm consists of an actor function and $\mu(s_t|\theta^\mu)$ a critic function $Q(s_t, a_t|\theta^Q)$. Due to its good performance on continuous control problems, it uses the gradient of the Q function relative to the action directly for policy training.

$$a_i = \mu(s_i \mid \theta^\mu) + \mathcal{N} \tag{5}$$

The behavior policy $\mu$ is a random process generated from the current online policy and random noise $OU.OU$ represents the value obtained by the Ornstein-Uhlenbeck from which the random process is sampled. $N \sim OU(\mu, \sigma^2)$ is a stochastic process that allows action exploration. This further noise exploration ensures that the agent behavior does not converge prematurely to a local optimum. The key advantage of our DDPG is that the exploration starting point can be better initialized by learning human expectations, which helps to significantly reduce the thorough exploration that can take days in the early stages of the DDPG. Starting from a better state, stochastic action exploration allows RL to further refine actions based on simulator feedback and produce more general and robust driving strategies.

Unlike the traditional random initialization $\theta^\mu$ of the DDPG, our DDPG is proposed to be initialized by simulating pre-trained $\theta^I$ loading as parameters $\theta^\mu$. In this paper, we define $s_t = \{o_t, f_t\}$ for each step, o, f is observed from a camera in the simulator, and F is additional obstacle perception information, i.e., the imitation learning phasedefined above.

When the number of samples in the replay buffer exceeds the batch size, it starts training the actor and critic network and optimizes them at each step based on Equations (7) and (9). Definition of loss for $Q$-networks: a similar approach to supervised learning, define the loss as MSE: mean squared error.

$$L = \frac{1}{N} \sum_i (y_i - Q(s_i, a_i | \theta^\rho))^2 \tag{6}$$

$$y_i = r_i + \gamma Q' \left( s_{i+1}, \mu' \left( s_{i+1} \mid \theta^{\mu'} \right) \mid \theta^{Q'} \right) \tag{7}$$

$y_i$ is calculated using the target strategy network $\mu'$ and the target $Q$ network $Q'$. This makes the learning of $Q$-network parameters more stable and converge. This label relies on the target network we are learning from, which is what distinguishes it from supervised learning.

Using the running average, the parameters of the online network are soft updated to the parameters of the target network.

$$\begin{aligned} \theta^{Q'} &\leftarrow \tau\theta^Q + (1-\tau)\theta^{Q'} \\ \theta^{\mu'} &\leftarrow \tau\theta^\mu + (1-\tau)\theta^{\mu'} \end{aligned} \tag{8}$$

On the other hand, the actor-network is further updated by a gradient descent step.

$$\nabla_{\theta^\mu} J_\beta(\mu) \approx \frac{1}{N} \sum_i \left( \nabla_a Q\left(s, a \mid \theta^Q\right) \Big|_{s=s_i, a=\mu(s_i)} \cdot \nabla_{\theta^\mu} \mu\left(s \mid \theta^\mu\right) \Big|_{s=s_i} \right) \tag{9}$$

Firstly, the simulation environment is initialized, the network parameters are set, and the network hyperparameters are passed to the Actor target network via the Actor-network, which is shown in Figure 6. The target network is mainly used to solve the target action, and the action is passed to the Critic target network, and the output is passed to the Critic network. As shown in Figure 7.

## Actor Network

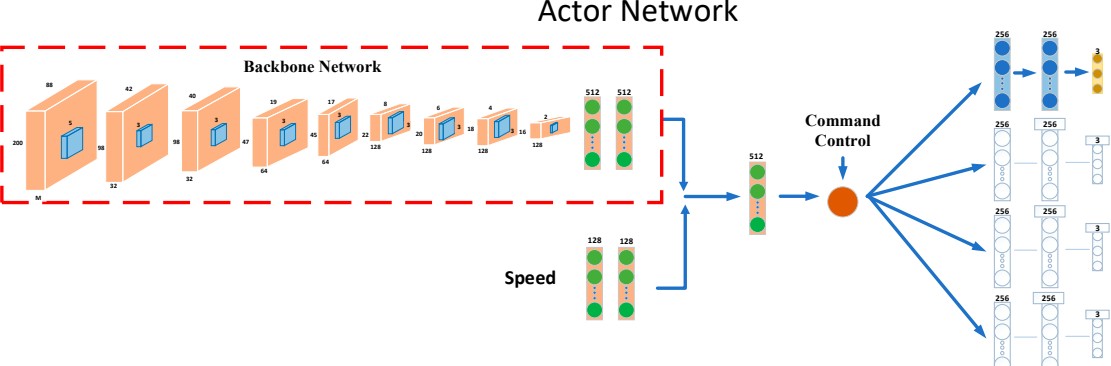

**Figure 6.** Actor Network.

## Critic Network

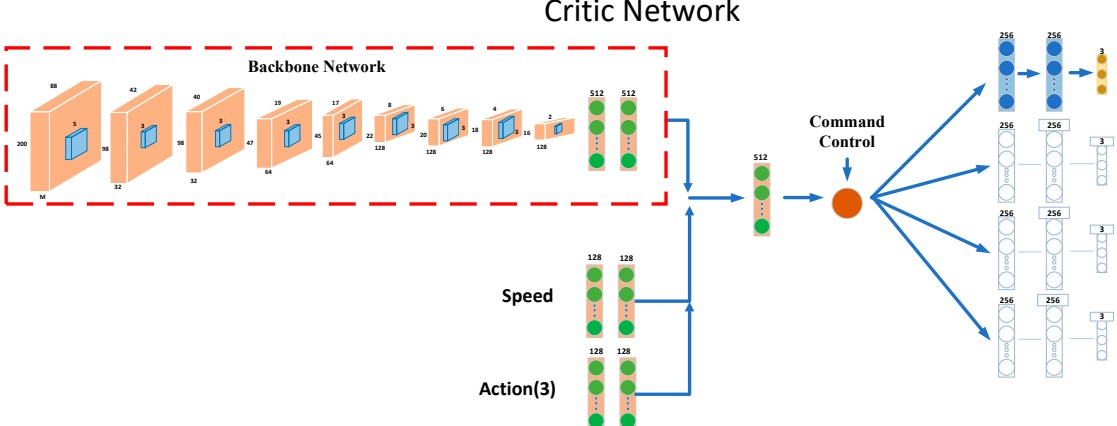

**Figure 7.** Critic network.

The input data information is also sampled for evaluation and passed to the Actor target network, the Critic target network, and the Critic network, respectively. The Critic network is then updated to pass the information to the Actor-network, generating an initial policy, which is explored and given the appropriate reward. When the reward is maximized, an optimal policy (action) is obtained, and this action is applied to the controller to drive the vehicle.

### 5.2. Reward Function of DDPG

The simulation environment is first initialized, and the parameters are set. The parameters are passed through the Actor network to the Actor target network, mainly used to solve the target actions, passing the actions to the Critic target network and the output return values to the Critic network.

At the same time, the input data information is sampled for evaluation and passed to the Actor target network, the Critic target network, and the Critic network, respectively. The Critic network is then updated to pass the information to the Actor-network, producing an initial policy.

The strategy is explored, and a reward is given. When the reward is maximum, an optimal strategy (action) is obtained, and the vehicle is driven by applying this action to the controller. The design is based on the following parameters: efficiency, safety, and comfort. For the DDPG algorithm, the design of the reward value function is important in influencing the network, guiding the direction of the gradient of parameter updates throughout the network. In the framework of reinforcement learning, the process by which an intelligence learns to adapt to its environment is guided by the reward function. A suitable reward function not only makes the strategies learned by the intelligence more reasonable but also allows the intelligence to learn faster and better the convergence of the network.

$$r_v(c) = \begin{cases} \min(25, v) & \text{if } c \text{ for Follow} \\ \min(35, v) & \text{if c for Straight} \\ v & \text{if } v < 20 \quad c \text{ for TurnLeft and TurnRight} \\ 40 - v & \text{if } v > 20 \quad c \text{ for TurnLeft and TurnRight} \end{cases} \tag{10}$$

$$r_s(c) = \begin{cases} 10 - |\delta| \times 20 & \text{if } c \text{ is straight} \\ -k_s \times \max((|\delta| - 0.2), 0) & \text{if } c \text{ is left or right} \end{cases} \tag{11}$$

In Equation (10), the reward parameter is a negative reward penalty for the vehicle speed dropping below the reference speed Vref, where the relationship between the negative reward and the vehicle speed is a one-time positive proportional relationship. When the vehicle speed reaches the reference speed, then it is rewarded positively. The advantage of this is that it increases the damping of speed changes and prevents the excessive pursuit of rewards from causing speed changes and uneven driving.

In Equation (11), we want the steering angle to be 0 when the vehicle is going straight, and we give a larger penalty when this value is larger. When turning left or right, it is desired that the vehicle's steering angle can be smoother when the intelligent body acts on corner crossing, obstacle avoidance, etc. Therefore, the magnitude of the steering wheel turning angle $\delta$ is considered. where $k_s$ is the penalty factor for vehicle directional cornering and the output range for steering wheel cornering is $[-1, 1]$.

Finally, both are set to $-100$, overlapping with the pavement and opposite lane. Collision damage is $-100$ for collisions with other vehicles and pedestrians and $r_d$ $-50$ for other objects (e.g., trees and utility poles).

The magnitude of the steering wheel corner of the vehicle: This reward parameter is a penalty for the vehicle to make a large hitting corner. It is hoped that the vehicle's cornering can be smoother when the intelligent body acts by crossing curves and avoiding obstacles, so the magnitude of the steering wheel corner $\delta$ is considered.

Where is the penalty factor for the vehicle directional turn, the output range for steering wheel turn is $[-1, 1]$ and $R_s$ is the vehicle directional turn reward term. The final reward r conditions for different command controls are calculated as follows.

In summary, the final reward function can be obtained as follows:

$$r = R(o, a) = r_s(c) + r_v(c) + r_r + r_o + r_d \tag{12}$$

## 6. Simulation Experiments

### 6.1. Experimental Settings

This paper uses the CARLA simulator [20], which simulates the urban environment with high fidelity CARLA environment contains dynamic obstacles such as self-cars, pedestrians crossing the road randomly, etc. The CARLA simulator provides 14 weather conditions, GPS, sensory measurements, and a rough plan consisting of coarse waypoint coordinates in a map without any fine-grained trajectory reference.

We pre-trained the actor network using the same experimental setup as in [27] to demonstrate the effectiveness of our imitative reinforcement learning. Fourteen hours of driving data collected from CARLA were used for training, and the network was trained using the Adam optimizer. In the imitative learning section, the setup details were the same as mentioned above. In the reinforcement learning section, the environment was dynamic by setting the number.

The maximum number of vehicles ranges from 20 to 40, and the number of pedestrians ranges from 50 to 100. Setting the maximum set to 1000 results in a maximum number of steps per turn of 3000. The remaining parameters used for model training are listed in the Table 3 below:

**Table 3.** Training assessment parameters.

| Parameters | Value |
|---|---|
| Buffer size | 100,000 |
| Batch size | 32 |
| Discount factor | 0.99 |
| Learning rate of actor | 0.0001 |
| Learning rate of critic | 0.001 |
| Max steps per episode | 3000 |
| Total training episodes | 8000 |
| Optimizer | Adam |
| Train_PLAY_MODE = 0 | Train = 1 Test = 0 |
| $\tau$ | 0.001 |

TRAIN_PLAY_MODE is the DDPG operating mode, when this parameter is set to 1 it enters training mode and when this parameter is set to 0 it enters test mode.

Where tau is the hyperparameter of the target network, lra is the learning rate of the actor-network or is the learning rate of the actor-network, buffer_size is the maximum capacity for storing experience samples, batch_size is the size of each BATCH acquisition, gamma = 0.99 the reward discount factor of the agent, the larger the discount factor, the more "long-term" the agent is thinking, the smaller the agent is more "immediate." "episodes_num is the number of training rounds the agent performs, max_steps is the maximum number of steps the agent can perform per round, and ACTION_NOISE is the number of exploration rounds the agent performs. NOISE is the coefficient by which the agent explores.

Several goal-directed tasks were evaluated using the CARLA benchmark [8], including "straight line," "one lap," "navigation," and "dynamic navigation."

The experiment is divided into three phases, each with a different level of difficulty, with an overall gradual increase in difficulty, and each phase of the task has two different weather environments, sunny and rainy. Each stage involved driving the vehicle with a destination as the goal, and the task focused on how well the vehicle performed in

straight-line driving, turning, and decision planning tasks in dynamic traffic, so the effects of signals and yellow lanes were not taken into account during the training test. The test map is Town04 in CARLA, a CARLA map of a city with eight lanes in a circle, with elevated, circular, uphill, and downhill scenarios, three-way intersections, etc.

In order to more fully consider the influence of weather on decision planning, this paper divides the weather into two groups, Weather1, and Weather2, referring to the work of Xiaodan Liang et al. [27]. Weather1 includes sunny days, sunny sunsets, rainy days, and days after rain. weather2 includes cloudy days and rainy days at sunset. The details are shown in Figure 8 below.

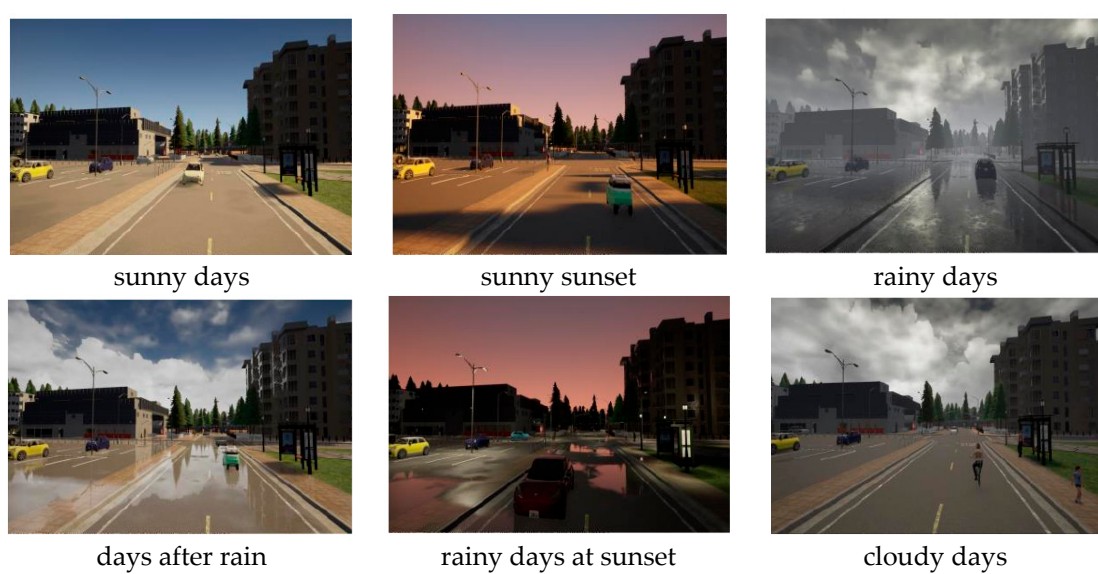

**Figure 8.** Weather conditions in different environments.

*6.2. Results*

In this experiment, we compare our method with the original MP from [3] and two state-of-the-art approaches: CIL [25] and CIRL. CIRL combines imitation learning with reinforcement learning. To evaluate the generalization performance in unknown weather conditions and environments, we take all four tasks for all methods in four driving conditions denoted as "Training condition," "New town", "New weather", and "New weather&town".

As shown in Figure 9, our model greatly outperforms the MP and CIL baseline tasks in almost all respects. It can be observed that the model achieves better performance than the others to some extent. Although the success rate of the model in this paper is, in some cases, inferior to that of CILR, this may be because the demonstration dataset used for training is smaller.

In addition, the model achieves better performance. Robustness and generalization in unknown environments while it takes less time to train. Our actor commentary network was trained using only 200,000 simulation steps driving non-stop at ten frames per second for about 8 h. This compares to about 12 h in CIRL, where it took 300,000. The results in "New weather&town" show that our model improves generalization performance, just like using large-scale demonstration data. Our model can obtain a high percentage of completed episodes after a few hours with good sampling efficiency, thanks to a good start of exploration driven by a controlled imitation phase. The proposed approach is implemented on the TensorFlow framework.

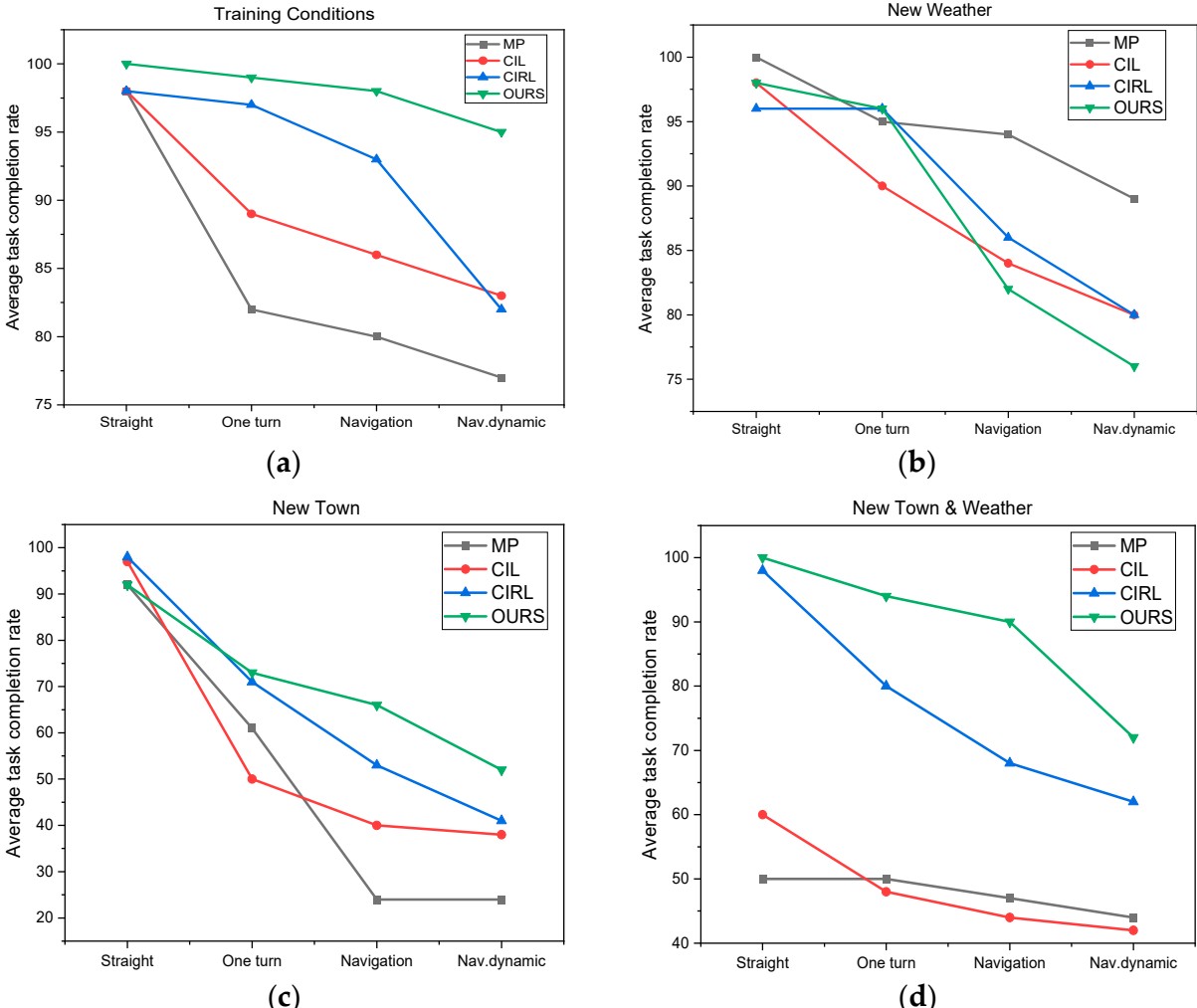

**Figure 9.** Comparison of the completion of the algorithm in this paper with the benchmark algorithm ((**a**) is in a training environment; (**b**) is in a new weather environment; (**c**) is in a new town environment; (**d**) is in a new weather and town environment).

As can be seen from Figure 10, the average score obtained by our algorithm is about 400. The higher the score, the better the driving condition of the algorithm in the experiment and the smaller the collision and violation ratio, which proves the effectiveness of the algorithm. The reward function 400 indicates that convergence has been achieved in the training results, i.e., that one can design a reward function according to their design and thus achieve the requirements of their design; 600 is the maximum value of the reward one can possibly achieve and is the reward reported for certain special driving situations.

This paper analyses the test results for straight ahead, cornering, and mixed conditions during vehicle driving, with a benchmark assessment test mainly for the mixed conditions.

From the results of the Tables 4 and 5, we can see that in the same urban environment, using the same weather conditions as in the training, the car did not have lane deviation and only just had the event of driving in other lanes, the average number of occurrences was 1.33; in the same urban environment and different weather conditions, the car also had better task completion, only had lane deviation and driving in other lanes violations, but Under the same weather conditions and different urban environments, the robustness of the model is also good, with a higher average task completion, a larger average distance travelled by the car before the violation occurred, and a lower number of violations; however, under different urban environments and weather conditions, the adaptability of the model decreases, and the probability of violation also relatively. Although the average

task completion is high, this is because the straight ahead working condition is relatively simple and the distance between the beginning and end of the task point is not very long, so the model is able to complete more of the tasks. In this condition, there are no dynamic objects, and the task distance is short, so there are no collisions, collisions with people or static collisions, etc. From the results of several other benchmark evaluations, the model in this paper can complete the task well in the straight-ahead condition.

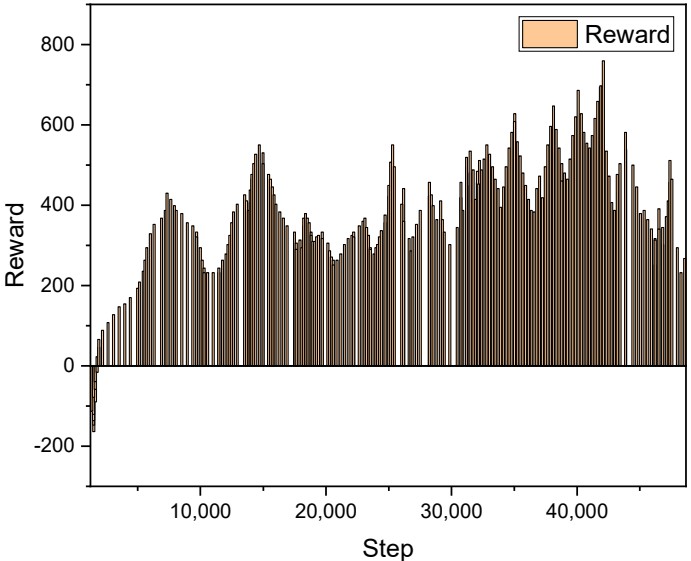

**Figure 10.** Average award (Value of reward function with step size).

**Table 4.** Straight: Average distance traveled.

| Town | Weather ID | Lane Departure | Driving in other Lanes | Crash | Hitting Someone |
|---|---|---|---|---|---|
| Town 1 | 1 | 16.74 | 14.13 | 16.74 | 16.74 |
|  | 2 | 11.58 | 12.05 | 13.26 | 13.26 |
| Town 2 | 1 | 9.41 | 9.54 | 11.95 | 11.95 |
|  | 2 | 13.37 | 12.79 | 15.46 | 15.46 |

**Table 5.** Straight: The average number of violations.

| Town | Weather ID | Lane Departure | Driving in other Lanes | Crash | Hitting Someone |
|---|---|---|---|---|---|
| Town 1 | 1 | 0 | 1.33 | 0 | 0 |
|  | 2 | 2.54 | 1.68 | 0 | 0 |
| Town 2 | 1 | 3.01 | 2.86 | 0 | 0 |
|  | 2 | 3.47 | 4.20 | 0 | 0 |

From the results of Tables 6 and 7, the car driving environment under turning conditions is relatively simple, and when the training conditions are the same, the model in this paper is basically able to drive through the whole section of the road, and the number of collisions is also low; in other weather conditions there is a slight decline, but due to the existence of advanced control commands, the model in this paper is less sensitive to the weather when the weather changes, and it can complete the driving task excellently. Adaptation ability decreases significantly, a part of the task volume will be lost, and the number of collisions also increases significantly, which is because the change in the urban environment will cause the structure of the image to change, and the algorithm will not

extract enough information about the features with greater recognition degree, which affects the neural network's judgment of the recognition results.

**Table 6.** One turn: Average distance traveled.

| Town | Weather ID | Lane Departure | Driving in other Lanes | Crash | Static Collision |
|------|------------|----------------|------------------------|-------|------------------|
| Town | 1 | 21.09 | 20.21 | 22.58 | 22.58 |
| 1 | 2 | 22.26 | 21.54 | 26.71 | 26.71 |
| Town | 1 | 16.21 | 17.26 | 21.39 | 21.39 |
| 2 | 2 | 16.45 | 14.36 | 24.83 | 24.83 |

**Table 7.** One turn: The average number of violations.

| Town | Weather ID | Lane Departure | Driving in other Lanes | Crash | Static Collision |
|------|------------|----------------|------------------------|-------|------------------|
| Town | 1 | 1.14 | 2.02 | 0 | 0 |
| 1 | 2 | 2.47 | 3.63 | 0 | 0 |
| Town | 1 | 4.33 | 4.67 | 0 | 0 |
| 2 | 2 | 6.68 | 6.97 | 0 | 0 |

From the results of Tables 8 and 9, although the overall performance of the model in this paper is not as good as that of the individual working conditions, the vehicle can complete the driving task well under the same conditions as the training, which is already a good result in the long-distance driving task. The model is robust and generalizable. In the same urban environment and under different weather conditions, the generalization ability of the model in this paper decreases, but it can still perform the driving task well. The average driving distance before a violation is less than that in Weather ID = 1, the number of violations increases, and the probability of collision increases, but the test results are relatively good, which shows that the algorithm is not particularly sensitive to changes in weather. The algorithm is not particularly sensitive to changes in weather, and under the same test environment, the model can still have good stability and prediction ability. However, when the algorithm was transferred to an unfamiliar urban environment, the model was unable to complete the test task properly in Town02 due to the longer coordinate distances of the task points and the increased dynamic factors in the environment, which resulted in more violations and more frequent collisions when the vehicles were driving, shorter normal driving distances and a significant decrease in driving effectiveness. This may be due to the dynamic and unstructured nature of the new urban environment and the limited feature information extracted by the network, resulting in an inability to optimize the representational capability of the network and a reduction in the adaptability of the model.

**Table 8.** Navigation in dynamic: Average distance traveled.

| Town | Weather ID | Lane Departure | Driving in other Lanes | Crash | Hitting Someone |
|------|------------|----------------|------------------------|-------|-----------------|
| Town | 1 | 22.91 | 20.27 | 23.69 | 21.54 |
| 1 | 2 | 19.95 | 17.71 | 19.81 | 20.23 |
| Town | 1 | 8.67 | 10.26 | 12.94 | 10.37 |
| 2 | 2 | 5.26 | 7.68 | 8.34 | 10.26 |

**Table 9.** Navigation in dynamic: The average number of violations.

| Town | Weather ID | Lane Departure | Driving in other Lanes | Crash | Hitting Someone |
|------|------------|----------------|------------------------|-------|-----------------|
| Town | 1 | 2.82 | 3.58 | 3.12 | 3.33 |
| 1 | 2 | 3.85 | 5.33 | 4.62 | 3.14 |
| Town | 1 | 11.0 | 9.34 | 7.67 | 6.41 |
| 2 | 2 | 12.18 | 13.68 | 10.73 | 9.13 |

## 7. Conclusions

In this paper, a two-stage framework is proposed to address the challenges of autonomous urban driving in complex environments and adverse weather conditions. This paper combines reinforcement learning with imitation learning. Extensive experiments conducted on the CARLA simulator show that the present model significantly improves robustness and generalization performance under a variety of driving conditions.

While the results are admirable, there is also significant scope for improvement under more challenging driving conditions. While the driving agent trained by the CIL-DDPG method learns reasonably good driving strategies for navigation tasks in dynamic environments, such as slowing down to avoid car collisions, there are more robust driving strategies that the agent needs to learn, such as obeying traffic rules and avoiding collisions with pedestrians.

With the continuous development of artificial intelligence applications and autonomous driving technology, urban autonomous driving technology has an extremely important role to play in the safety of vehicles and the efficiency of traffic travel. In this paper, an end-to-end approach based on conditional imitation learning is proposed using the idea of multi-input neural networks and validated by simulation with the CARLA simulator to demonstrate the effectiveness of the algorithm and provide a feasible solution for the study of autonomous driving technology.

**Author Contributions:** Methodology, L.H. and G.D.; software, L.H.; resources, J.O.; writing—original draft preparation, L.H. and M.B.; writing—review and editing, E.Y.; project administration, J.O.; funding acquisition, J.O. All authors have read and agreed to the published version of the manuscript.

**Funding:** Thanks to the sponsorship provided by Special key project of technological innovation and application development of Chongqing (cstc2020jscx-dxwtBX0048) and Science and Technology Research Project of Chongqing Municipal Education Commission (KJQN201901146).

**Institutional Review Board Statement:** This study waived ethical review and approval, because it does not involve humans or animals.

**Informed Consent Statement:** Not applicable.

**Data Availability Statement:** As the project involves confidentiality, research data is not provided. If readers need research data, please contact the corresponding author.

**Conflicts of Interest:** The authors declare no conflict of interest.

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
