# Peer review of "Imitative Reinforcement Learning Fusing Mask R-CNN Perception Algorithms"

_applsci, doi:10.3390/app122211821_

Round 1

Reviewer 1 Report

1. What is DDPG in the abstract?

2. In the abstract, be consistent with Carla or CARLA.

3. The tenses in your abstract are also contradicting.  First sentences use present tense while last statement uses past tense.

4. In the last statement of the abstract, "unknow environments" should be clarified.  Give an exact example since you already have done this in the simulation.

5. In paragraph 2 of Introduction.  You introduced reinforcement learning and presented its weakness.  Place proper citations here.  Add also discussion about the weaknesses of reinforcement learning.

6. Please provide examples of challenging navigation tasks.

7. Figure 1 is not a flow chart.  Also please place an adequate caption for Figure 1.  I do not understand the arrow from the imitation stage to the reinforcement block.  Where is exactly the input coming from, the obstacle classification?  What does "Share Weighs" mean?

8. Line 77 does not mean anything.

9. Line 79 --> It should be "innovations" and "contributions".

10. Contribution 1 should be rewritten.  Focus on what makes CIL-DDPG different when using RL and IL.  This contribution lacks details.

11. A new reward function is introduced in contribution 2.  What is the innovation here?  Again, lack of details.

12. Extensive simulation should be another contribution.  Describe your simulation scenarios and results compared to benchmarks.

13. In the outline section, remove Section I.

14. In the introduction, the problem statement is unclear particularly what are the weaknesses being solved.  This was brought by lack of given examples and related literature trying to overcome the problem.  You mentioned also that you want to solve sever weather conditions.  But, were these also the problems being addressed by previous works?

15. In your related works, why is there no literature discussing severe weather conditions?  This topic has already been tackled.  Please include this in the RRL.

16. Why is the title for Section 3 and 3.1 the same?

17. Is # of vehicles = 15 a considerable number in the experiment?  Please justify your choices in the dataset acquisition.  Are these different or better than previous studies?

18. What type of images were captured in the data acquisition part?  Please specify.

19. In Table 1, specify what the description really means, e.g., -1.0 means full left, 1.0 full right, etc.

20. Figure 3 can be presented in a way how the image enhancement has been discussed.

21. Why is Figure 4 above the figure, while the others are below the figures?

22. There is a Figure 5, but, there is no discussion for it or it is not referenced in your text.

23. The font sizes of your equations are varying in sizes.  Equation number labels are also misplaced.

24. The captions of your figures are lacking in discussions and details.  Please provide a better caption. 

25. The reward function must be clearly explained.  Honestly, this is not a novel contribution since the min and max functions are simple to use.  I ask the authors to explain clearly the arguments in their reward functions.  How were they chosen.  Isn't this more like a fuzzy control?  How were the choices relevant to the severe weather conditions in your experiments?

26. In Figure 8, clear descriptions of Weather 1 and Weather 2.  Why are there three figures in each weather?  Also, I thought these were severe weather conditions?  How come?

27. In Figure 10, what does a reward equal to 400 or 600 mean?  What is its physical meaning?

Author Response

Ans1:The Deep Deterministic Policy Gradient (DDPG) algorithm is an online-based deep reinforcement learning algorithm proposed by the DeepMind team for solving continuous control problems.

Ans2:All carla are unified as CARLA.

Ans3:All the tenses of the summary are in the general present tense and the rest are passive statements.

Ans4:The so-called unknown environment, Carla's Twon04 map used in this paper, generates 20 vehicles at random, selects different weather conditions and performs tasks with dynamic traffic scenarios, one straight ahead and one turning task.

Ans5: In paragraph 2 of the Introduction, the weaknesses of reinforcement learning are introduced. The weaknesses are discussed.

Ans6:Challenging navigation tasks will be added later in the dynamic experiment.

Ans7:Firstly the title of Figure 1 has been amended. Secondly the classification of obstacles is a function of mask_rcnn and the input is the camera from the car; finally, the shared weights indicate that I am training the model for good imitation learning and then using the control

input for reinforcement learning.

Ans8:Line 77 has been deleted.

Ans9:Line 79 has been corrected

Ans10:Contribution 1 is explained in detail.

Ans11:The design of the reward function itself was designed to meet the relevant experimental requirements of this paper, which in itself is a unique design and is somewhat innovative.

Ans12:In the results section of the experiment, the results of this paper's methodology compared to the benchmark are discussed.

Ans13: In the outline section, section I has been deleted.

Ans14:In the introduction section add a discussion of the problem to be solved in this paper and how to go about solving it, in addition to a solution to the severe weather I have to carry out.

Ans15:In addition to this, the paper will also consider bad weather, for example, relevant references have been added and discussed

Ans16:the title for Section 3 and 3.1 has been corrected.

Ans17:Firstly, 15 vehicles is not a maximum number, 15 is just a conventional number selection for this study; secondly the selection of this dataset allows for information on measured values, one of the biggest differences from the general dataset.

Ans18:In section 3.3, the acquired images are described in detail.

Ans19:Description of details added.

Ans20:The enhancement has been demonstrated.

Ans21:Figure 4 has been modified and placed below.

Ans22:Section 4.2 is the introduction to Figure 5.

Ans23:The formula size has been modified and right-aligned.

Ans24:The title of the chart and a more detailed description are given.

Ans25:In addition, the reward function will be discussed and introduced in detail in this paper.

Ans26: The setting of each weather was my reference to that Stanford paper. Also, you can think of it as a bad weather situation, which is what I really set up in my experiments.

Ans27:The reward function 400 clearly indicates that I have achieved convergence in my training results, i.e. that I can design my reward function according to my design and thus achieve the requirements of my design; 600 is the maximum value of the reward I can possibly achieve and is the reward reported for certain special driving situations.

Reviewer 2 Report

Title: Imitative Reinforcement Learning Fusing Mask R-CNN Perception Algorithms

In this work, an end-to-end autonomous driving approach is proposed. The proposed approach combines Conditional Imitation Learning (CIL), Mask R-CNN with DDPG. In the first stage, data acquisition is first performed by using Carla, a high-fidelity simulation software. Data collected by Carla is used to train the Mask R-CNN network. The segmented images are transformed into the backbone of CIL to perform supervised Imitation Learning (IL). DDPG means using Reinforcement Learning for further training in the second stage, which shares the learned weights from the pre-trained CIL model.

1- The abstract should contain results in terms of improvement ratio between the proposed and existing works.

2- Please fix grammatical errors. For example “this paper proposes a two-stage framework, called CIL-DDPG, is proposed which combines reinforcement learning …”. The authors need to check the entire manuscript for such typos and grammatical errors .

3- Some reference need to be included in the manuscript such as: 1) doi: 10.1109/TCSVT.2019.2905881, and 2) doi: 10.1109/DeSE54285.2021.9719469 .

4- between lines 88 and 93, Section I should be Section 1, Section II should read Section 2, and Section III should be Section 3.

5- A summary table should be included at the end of the related work section. The summary table should include: a) algorithm name and reference, b) brief methodology, 3) highlights, and 4) limitations.

6- Pseudo code for the 3.1 to 3.2 should be included to make it more clear to the readers .

7-Any equation, data, or figure should be cited with its reliable source if it is taken from a reference.

8- More related works can be added with providing some discussion about their drawbacks that are considered in this paper.

9- Figure 8 should be enlarged. Also, include more visual examples.

10- For Figures 9, a) include grid lines to increase the readability, b) arrange the subplots to be 2x2 and include sub-figure number (a, b, …).

11 – Results should be discussed thoroughly.

Author Response

Ans1:Summary added to the results presentation.

Ans2:The author has made corrections and checked the entire manuscript for typos and grammatical errors.

Ans3:References have been added doi.

Ans4: Between lines 88 and 93, the question about sections has been amended.

Ans5:A summary table has been added at the end of the relevant work section.

Ans6:Pseudo code has been added

Ans7:Any equations, data or figures that have been taken from references have been credited to their reliable source.

Ans8:This article adds more discussion of related work as well as some references to provide some discussion of their shortcomings.

Ans9:Figure 8 has been enlarged to increase readability, but Figure 8 is a simple weather change and I don't think too many examples are needed

Ans10:For Figure 9, changes have been made to increase readability

Ans11:In 6.2, the results section has been thoroughly discussed.

Round 2

Reviewer 1 Report

My comments have been addressed.

Author Response

All issues have been resolved

Reviewer 2 Report

Title: Imitative Reinforcement Learning Fusing Mask R-CNN Perception Algorithms

In this work, an end-to-end autonomous driving approach is proposed. The proposed approach combines Conditional Imitation Learning (CIL), Mask R-CNN with DDPG. In the first stage, data acquisition is first performed by using Carla, a high-fidelity simulation software. Data collected by Carla is used to train the Mask R-CNN network. The segmented images are transformed into the backbone of CIL to perform supervised Imitation Learning (IL). DDPG means using Reinforcement Learning for further training in the second stage, which shares the learned weights from the pre-trained CIL model.

The authors have addressed some of the raised comment; the following comments need to be addressed carefully :

1- The abstract has typos. For example, “comped” --> “compared”.

2 – The improvement ratio should be with the best benchmark .

3- Some reference need to be included in the manuscript such as: 1) doi: 10.1109/TCSVT.2019.2905881 (“Small Object Detection in Unmanned Aerial Vehicle Images Using Feature Fusion and Scaling-Based Single Shot Detector With Spatial Context Analysis”), and 2) doi: 10.1109/DeSE54285.2021.9719469 (“Object Detection and Distance Measurement Using AI”).

4- In line 81, “Related issues to be addressed in this paper:” should read “The related issues to be addressed in this paper are as follows:”

6- The included Pseudo code is very simple; the authors need to include a more detailed pseudo code to make it more clear to the readers to replicate the work presented in this paper.

Author Response

Ans1:The typo in the summary has been corrected.

Ans2: Improvement rates were compared not only to the best benchmark but also to other benchmarks.

Ans3:These two pieces of reference, as well as the additions, are numbered 6, 7.

Ans4: “Related issues to be addressed in this paper:” has been corrected as shown on line 95 “The related issues to be addressed in this paper are as follows:”

Ans6:A more detailed pseudo-code is shown